# INVESTIGATING INTRA-ABSTRACTION POLICIES FOR NON-EXACT ABSTRACTION ALGORITHMS

## ABSTRACT

One weakness of Monte Carlo Tree Search (MCTS) is its sample efficiency which can be addressed by building and using state and/or action abstractions in parallel to the tree search such that information can be shared among nodes of the same layer. The primary usage of abstractions for MCTS is to enhance the Upper Confidence Bound (UCB) value during the tree policy by aggregating visits and returns of an abstract node. However, this direct usage of abstractions does not take the case into account where multiple actions with the same parent might be in the same abstract node, as these would then all have the same UCB value, thus requiring a tiebreak rule. In state-of-the-art abstraction algorithms such as pruned On the Go Abstractions (pruned OGA), this case has not been noticed, and a random tiebreak rule was implicitly chosen. In this paper, we propose and empirically evaluate several alternative intra-abstraction policies, several of which outperform the random policy across a majority of environments and parameter settings.

## 1 INTRODUCTION

A plethora of important problems can be viewed as sequential decision-making tasks such as autonomous driving (Liu et al., 2021), energy grid optimization (Sogabe et al., 2018), financial portfolio management (Birge, 2007), or playing video games (Silver et al., 2016). Though arguably state-of-the-art on such decision-making tasks is achieved using machine learning (ML) as demonstrated by DeepMind with their AlphaGo agent for Go (Silver et al., 2016) or OpenAI Five for Dota 2 (Berner et al., 2019), there is still a demand for general domain-knowledge independent, on-the-go-applicable planning methods, properties which ML-based approaches usually lack but which are satisfied by Monte Carlo Tree Search (Browne et al., 2012) (MCTS), the method of interest for this paper. For example, Game Studios rarely implement ML agents as they have to be costly retrained whenever the game and its rules and updated. Though not within the scope of this paper, improvements to MCTS might also potentially translate to ML-based methods that use MCTS as their underlying search.

One research area to improve MCTS is using abstractions that aim at reducing the search space by grouping states and actions in the current MCTS search tree to enable an intra-layer information flow (Jiang et al., 2014; Anand et al., 2015; 2016), by averaging the visits and returns of all abstract action nodes in the same abstract node used for the Upper Confidence Bounds (UCB) formula in the tree policy. Inevitably, there are action nodes with the same parent state and same abstract node, which results in multiple actions having the exact same UCB value during the tree policy. Without giving this case special treatment, state-of-the-art algorithms like pruned On the Go Abstractions (pruned OGA) (Anand et al., 2016) simply perform tiebreaking exactly as in the non-abstracted case. In the case of pruned OGA, this is done randomly.

In this paper, we aim to tackle exactly this problem by proposing and evaluating several random-policy alternatives, several of which significantly enhance OGA's performance across a variety of environments and parameter settings. The contributions of this paper can be summarized as follows:

- We propose the **Alternating State And State-Action Pair Abstractions (ASASAP)** framework, which generalizes the abstractions built by most MCTS-based abstraction algorithms, including Automatic State abstractions (AS), Abstractions of State-Action Pairs (ASAP), and OGA.

- We empirically show that the case of having multiple abstracted actions with the same parent is not an edge case and occurs frequently.
- We propose and evaluate seven intra-abstraction policies as alternatives to the random policy, namely: **UCT**, **FIRST**, **GREEDY**, **MOST_VISITS**, **LEAST_VISITS**, **LEAST_OUTCOMES**, and **RANDOM_GREEDY**. The former, UCT, performs best overall, is a parameter-free drop-in improvement to OGA, as it performs either equally well or better across a wide range of parameters and problem settings. Furthermore, it causes only a negligible runtime overhead (see Tab. 4).

The paper is structured as follows. Firstly, in **Section** 2, we lay the theoretical groundwork for this paper. In particular, we define the ASASAP framework, which helps us introduce and classify other abstraction frameworks such as ASAP or AS from the literature. Next, in **Section** 3, we reiterate the intra-abstraction policy problem, describe seven alternatives, and illustrate on a concrete game tree how one of these modifications, using UCT as the intra-abstraction policy, can provably improve the performance. We then describe our experiment setup in **Section** 4. The experimental results are presented and discussed in **Section** 5, where we first measure the number of times an intra-abstraction policy has to be queried in the first place, followed by a thorough analysis of all proposed intra-abstraction policies with a focus on UCT. At the end, in **Section** 6, we briefly summarise our findings and provide an outlook for future work.

## 2 FOUNDATIONS OF NON-LEARNED DOMAIN-INDEPENDENT ABSTRACTIONS

**Problem model and optimization objective:** We use finite MDPs (Sutton & Barto, 2018) as the model for sequential, perfect-information decision-making tasks. Here, $\Delta(X)$ denotes the probability simplex of a finite, non-empty set $X$ and $\mathcal{P}(X)$ denotes the power set of $X$.

*Definition:* An *MDP* is a 6-tuple $(S, \mu_0, \mathbb{A}, \mathbb{P}, R, T)$ where the components are as follows:

- $S \neq \emptyset$ is the finite set of states, $T \subseteq S$ is the (possibly empty) set of terminal states, and $\mu_0 \in \Delta(S)$ is the probability distribution for the initial state.
- $\mathbb{A} \colon S \mapsto A$ maps each state $s$ to the available actions $\emptyset \neq \mathbb{A}(s) \subseteq A$ at state $s$ where $|A| < \infty$.
- $\mathbb{P} \colon S \times A \mapsto \Delta(S)$ is the stochastic transition function where we use $\mathbb{P}(s' \mid s, a)$ to denote the probability of transitioning from $s \in S$ to $s' \in S$ after taking action $a \in \mathbb{A}(s)$ in $s$.
- $R \colon S \times A \mapsto \mathbb{R}$ is the reward function.

From hereon, let $M = (S, \mu_0, \mathbb{A}, \mathbb{P}, R, T)$ be an MDP. Using the same notation as Anand et al. (2015), we also define $P := \{(s, a) \mid s \in S, a \in \mathbb{A}(s)\}$ as the set of all state-action pairs. The goal is to find an agent $\pi$ that we model as a mapping from states to action distributions $\pi \colon S \mapsto \Delta(A)$ such that $\pi$ maximizes the expected episode's return, where the (discounted) return for of episode $s_0, a_0, r_0, \ldots, s_n, a_n, r_n, s_{n+1}$ with $s_{n+1} \in T$ is given by $\gamma^0 r_0 + \ldots + \gamma^n r_n$.

**Abstraction frameworks** Next, we will define a general abstraction framework that includes most of the here-presented abstraction algorithms and captures their core working principle. We bluntly call this framework **A**lternating **S**tate **A**nd **S**tate-**A**ction-**P**air **A**bstractions (ASASAP) whose purpose is to unify parts of the abstraction zoo. The idea of ASASAP is to alternately construct a state abstraction from a state-action-pair abstraction and vice versa. For our purposes, we simply define state and action abstractions as equivalence relations (equivalently partitions) of the state or action space. In the supplementary materials in Section A.3, we show a concrete example of how an ASAP abstraction (a special case of ASASAP) is built.

*Definition*: We call the equivalence relation $\mathcal{H} \subseteq P \times P$ induced by some $n \in \mathbb{N}$, some initial state equivalence relation $\mathcal{E}_0 \subseteq S \times S$, mappings $f \colon \mathcal{P}(S \times S) \mapsto \mathcal{P}(P \times P)$ and $g \colon \mathcal{P}(P \times P) \mapsto \mathcal{P}(S \times S)$ to equivalence relations an $ASASAP_{f,g,n,\mathcal{E}_0}$ abstraction if it is of the form

$$\mathcal{H} = \mathcal{H}_n, \tag{1}$$

$$\mathcal{H}_{i+1} = f(\mathcal{E}_i) \qquad \forall i, \tag{2}$$

$$\mathcal{E}_{i+1} = g(\mathcal{H}_{i+1}) \qquad \forall i. \tag{3}$$

If additionally $\mathcal{H}$ is invariant to any number of additional applications of $f$ and $g$, then we call it *converged*.

Next, we will present some concrete instances of ASASAP from the literature. Firstly, Jiang et al. (2014) used- and Givan et al. (2003) proposed AS-UCT (the name was given by Anand et al. (2015)), which defines $g_{\text{AS}}(\mathcal{H}_{i+1})$ as grouping states if and only if they have identical legal actions and they are pairwise equivalent:

$$
\begin{aligned}
(s_1, s_2) \in g(\mathcal{H}_{i+1}) &\iff \mathbb{A}(s_1) = \mathbb{A}(s_2) \,\wedge \\
\forall a_1 \in \mathbb{A}(s_1) &: ((s_1, a_1), (s_2, a_1)) \in \mathcal{H}_{i+1}.
\end{aligned}
\tag{4}
$$

And any state-action-pair $(s_1, a_1), (s_2, a_2)$ is equivalent i.e. $((s_1, a_1), (s_2, a_2)) \in f_{AS}(\mathcal{E}_i)$ if and only if the state-action pairs have similar immediate rewards and transition distributions:

$$
|R(s_1, a_1) - R(s_2, a_2)| \leq \varepsilon_{\text{a}}
$$
$$
\text{and } F := \sum_{x \in \mathcal{X}} \left| \sum_{s' \in x} \mathbb{P}(s' \mid s_1, a_1) - \mathbb{P}(s' \mid s_2, a_2) \right| \leq \varepsilon_{\text{t}},
\tag{5}
$$

where $\mathcal{X}$ are the equivalence classes of $\mathcal{E}_i$ and $\varepsilon_{\text{t}}, \varepsilon_{\text{a}} \geq 0$. In general, for $\varepsilon_{\text{t}}, \varepsilon_{\text{a}} > 0$, $f_{AS}(\mathcal{E}_i)$ is not an equivalence relation because transitivity is not guaranteed. Hence, any abstraction algorithms using these need to slightly modify $f_{AS}(\mathcal{E}_i)$ to obtain an equivalence relation. The reason for allowing $\varepsilon_{\text{a}}$ and $\varepsilon_{\text{t}}$ to be greater than 0, is to find more correct abstractions at the cost of potentially abstracting state-action-pairs or states that do not have the same value. The experiments of this paper confirm that this can be beneficial.

To allow for the detection of more symmetries, Anand et al. (2015) proposed ASAP abstractions that are based on Ravindran & Barto (2004) homomorphism condition that does not require there to be a 1-to-1 action match but only a mapping of actions to each other, concretely $g_{\text{ASAP}}(\mathcal{H}_{i+1})$ is defined as

$$
\begin{aligned}
(s_1, s_2) \in g_{\text{ASAP}}(\mathcal{H}_{i+1}) &\iff \\
\forall a_1 \in \mathbb{A}(s_1)\, \exists a_2 \in \mathbb{A}(s_2) &: ((s_1, a_1), (s_2, a_2)) \in \mathcal{H}_{i+1} \\
\forall a_2 \in \mathbb{A}(s_2)\, \exists a_1 \in \mathbb{A}(s_1) &: ((s_1, a_1), (s_2, a_2)) \in \mathcal{H}_{i+1}.
\end{aligned}
\tag{6}
$$

The action abstraction $f_{\text{ASAP}}$ is the same as the previously defined $f_{AS}$ using $\varepsilon_{\text{t}} = \varepsilon_e = 0$, however, as we will later see there is nothing that would prevent us from choosing epsilon values greater than zero here.

**Abstractions for search:** Constructing ASAP or AS abstractions until convergence for an entire MDP is oftentimes infeasible, and such a computation would significantly hamper the runtime. Hence, ASAP-UCT (Anand et al., 2015), AS-UCT (Anand et al., 2015; Jiang et al., 2014), and OGA-UCT (Anand et al., 2016) build their ASASAP abstraction on the **local-layered MDP** rooted at the state $s_d$ where the decision has to be made.

*Definition:* The state space of the *layered MDP* of $M$ is $S \times \{0, \dots, h\}$ where $h \in \mathbb{N}$ is the horizon and if $(s, n)$ is a successor state of $(s', n')$, then $n = n'+1$ and any initial state has $n = 0$. Additional terminal states are $S \times \{h\}$. The *local-layered MDP* rooted at $s_d$ is the layered MDP of $M$ but with its states, actions, and possible state-action-pair-successors restricted to those present in the current search graph.

In local-layered MDPs, a converged ASAP or AS abstraction can be efficiently computed with dynamic programming, where one requires only the abstraction of the previous layer to compute the abstractions for the next. In ASAP-UCT and AS-UCT, an ASAP/AS-like abstraction is built in regular intervals on the current MCTS (for details on MCTS, see Section A.10) search graph using an initial state equivalence relation that groups all terminal states of the same layer, groups all non-fully-expanded nodes of the same layer, and puts all remaining nodes in their own abstract node of size one. The abstraction built by ASAP/AS-UCT differs only from the ASAP/AS abstraction in that non-fully-expanded nodes are never grouped with fully-expanded nodes. This non-grouping condition also applies to OGA (Anand et al., 2016). For the later experiments, we will also experiment with grouping partially explored state nodes as in ASAP-UCT, but for OGA. We refer to this parameter as PG $\in \{0, 1\}$ where 0 refers to no partial grouping.

Unlike ASAP-UCT and AS-UCT, the successor of ASAP-UCT, OGA-UCT, does not compute its respective abstraction from the ground up but rather attempts to approximate the ASAP abstraction by rebuilding only parts of its current abstraction. More concretely, OGA-UCT tests every $K$-th Q node visit if the abstraction needs to be updated (e.g., new successors were sampled that invalidate a previous abstraction). If so, the parent's abstraction is recursively updated too.

A core weakness of ASAP abstractions is their exactness, which causes them to not be able to deal with stochasticity well. Hence, Anand et al. (2016) directly proposed *pruned OGA* as an improvement to OGA-UCT, which is the same as OGA-UCT except that for the abstraction construction step for each state-action pair with $n$ successors with respective probabilities $p_1, \ldots, p_n$ only those with $p_i > \alpha \cdot \max\{p_1, \ldots, p_n\}$, $\alpha \in [0, 1]$ are considered. Also in this paper, we consider $(\varepsilon_a, \varepsilon_t)$-OGA (Schmöcker et al., 2025) which is equivalent to OGA-UCT except that one allows for $\varepsilon_a, \varepsilon_t$ to be greater than 0. Since this does not induce an equivalence relation, the abstraction construction process has to be slightly modified as detailed by Schmöcker et al. (2025).

**Abstraction usage:** Thus far, we have only discussed how to build abstractions but not how to use them. The key mechanism that all here-presented MCTS-based abstraction methods use (e.g. AS-UCT, ASAP-UCT, OGA-UCT) is only to modify the tree policy by enhancing the UCB value. The UCB value for an action is enhanced by using the aggregated visits and returns of all actions that are part of the same abstract action (i.e. equivalence class). In particular, state abstractions are not used at this stage. These are only needed as an intermediate step to find action abstractions. Only AS-UCT differs slightly from this approach as it only aggregates the statistics of actions that additionally have the same abstract parent. This is because AS-UCT was originally intended as a state only abstraction which is why it did not decouple action and state abstractions.

The intra-abstraction policies that we will later propose only affect the abstraction usage component of an abstraction algorithm. They do **not** modify the abstraction-building process itself.

**Other automatic abstraction algorithms:** A different abstraction paradigm is PARSS by Hostetler et al. (2015) that initially groups all successors of each state-action pair. As the search progresses, this coarse abstraction is refined by repeatedly splitting abstract nodes in half. Another technique is to build, but then fully abandon an abstraction mid-search, a method coined Elastic MCTS by Xu et al. (2023). Though not fully domain-independent, another approach is given by Sokota et al. (2021), who group states based on a domain-specific distance function, and the maximal grouping distance shrinks as the search progresses. While also not in scope of this paper, research effort on abstractions is also dedicated to continuous and/or partially observable problems (Hoerger et al., 2024), and learning-based methods, such as learning and planning on abstract models (Ozair et al., 2021; Kwak et al., 2024; Chitnis et al., 2020), or building abstractions that rely on learned functions (e.g. a value function) (Fu et al., 2023). Research effort has also been dedicated towards automatic abstractions of the transition function, which on an abstract level can be described as pruning certain successors from the transition function (Sokota et al., 2021; Yoon et al., 2008; 2007; Saisubramanian et al., 2017).

## 3 METHOD

**Intra-abstraction policies:** A consequence of ASAP's key idea to decouple state and action abstractions is that two state-action pairs may be abstracted even when they have the same parent. This, however, leads to the thus-far overlooked problem that any two abstract Q nodes with the same parent will have an identical UCB value (see Section A.10) as they have the same number of aggregated visits and returns. Hence, a tiebreaking rule is needed, which we refer to as an **intra-abstraction policy**. Anand et al. (2016) implicitly chose a random intra-abstraction policy. While this random policy causes no harm when the abstractions are lossless, when dealing with lossy abstractions (i.e., those where states or actions could be abstracted even when they do not have the same value under optimal play) a random policy can be detrimental to performance as we will show in the experiment section 5.

We propose a number of alternative intra-abstraction policies to choose an action within the selected abstract node. The intra-abstraction policy can be split into two phases. One for the decision policy (i.e. for the final decision at the root node) and one for the tree policy.

We separate the to-be-proposed methods into four groups. The first group encompasses the implicitly used methods from the literature. The second group includes policies that focus on exploration, the third group focuses on exploitation, and the fourth group is a mix of both.

**1. RANDOM**: Randomly choose an action with uniform probability for both the decision and tree policy. This is the one used by Anand et al. (2015)

**2. FIRST**: Simply choose the first encountered ground action for both the decision and tree policy. Though we do not believe that this method would perform well, this is one of the standard tiebreaking rules for MCTS and might be accidentally implicitly used in an abstraction algorithm.

**3. RANDOM_GREEDY**: Choose the action randomly during the tree policy and greedily during the decision policy. We include this method as an ablation to pin down the influence of simply being greedy in the decision policy.

**4. LEAST_VISITS**: Choose the action with the least number of visits with a random tiebreak if two actions have the same number of visits. Greedy policy as the decision policy. This policy is closest to RANDOM except that visits are distributed perfectly evenly.

**5. LEAST_OUTCOMES**: Choose the action with the minimal probability sum of all thus-far sampled successors. Greedy policy as the decision policy. The idea is to allow the detection of any faulty abstractions as soon as possible.

**6. GREEDY**: Choose the action with the highest Q value.

**7. MOST_VISITS**: Choose the action with the most visits. Greedy policy is the decision policy. The idea behind this policy is to increase the search depth, because when the abstraction does not change, the action that wins the first tiebreak will always be chosen, as it will be the only one receiving visits. However, this may come at the cost of exploration.

**8. UCT**: Choose the action whose ground visits and values maximize the UCB value (see Section A.10) using the same exploration constant as the UCB selection for the abstract action. Greedy policy is the decision policy.

All policies use a random tiebreak e.g., when two ground actions have the same number of visits when using the LEAST_VISITS policy.

**Case study RANDOM vs UCT:** Next, we are going to study the theoretical properties of the RANDOM versus the UCT intra abstraction policy. Firstly, given arbitrary abstractions, no guarantees that go beyond those of any OGA-based methods can be made, as one can always construct abstractions such that intra-abstraction policies would never be queried. However, if we assume a special case of abstractions, which are those that only group state-action pairs with the same parent node, then guarantees can be made. Firstly, RANDOM in combination with an arbitrary same-parent state-action pair abstraction is not guaranteed to converge to the optimal action. An example where RANDOM fails to find the optimal action is given in Fig. 1. In contrast, using UCT will always converge to the optimal action. Concretely, assume a decision has to be made at state $s_\mathrm{d}$.

*Theorem 1*: Let $\mathcal{E}$ be a same-parent state-action pair abstraction of the local-layered MDP rooted at $s_\mathrm{d}$. Consider MCTS that uses the aggregated abstract visits and returns of $\mathcal{E}$ for the UCB value calculation for any state-action pair in combination with the UCT intra-abstraction policy. This MCTS version's ratio of root node visits of the optimal action(s) to the number of iterations will converge in probability to 1.

The proof of this theorem is provided in the supplementary materials in Section A.1. A direct consequence of this theorem is that pruned OGA's or $(\varepsilon_\mathrm{a}, \varepsilon_\mathrm{t})$-OGA's root visits will also, in probability, converge to the optimal action(s) if they were slightly modified to only group state-action pairs with the same parent. This is because eventually their abstractions will converge since all the MDP's state-action pairs will be visited and all their outcomes will be sampled almost surely, which allows only to apply Theorem 1.

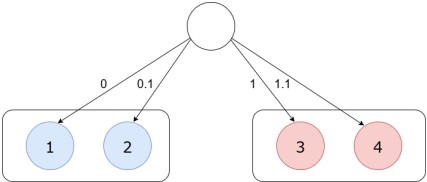

Figure 1: Assume that MCTS with the visualized fixed action abstraction is run on the following deterministic depth-1 game tree. This abstraction would also be discovered by $(0.1, 0)$-OGA since when all four actions are played at least $K$ times, $(0.1, 0)$-OGA will have abstracted actions 1,2, and 3,4. While the visits will converge to choosing the abstract node with actions 3 and 4 for both the RANDOM and UCT intra-abstraction policy, RANDOM will distribute its visits uniformly amongst 3 and 4, resulting in an average payoff of 1.05. This shows that with RANDOM, convergence to the optimal action is in general not guaranteed. In contrast, the UCT intra-abstraction policy guarantees convergence to the average payoff of 1.1 by converging to action 4.

## 4 EXPERIMENT SETUP

In this section, we describe the general experiment setup. Any deviations from this setup will be explicitly mentioned.

**Parameters:** Originally, OGA-UCT (Anand et al., 2016) used the absolute value of the abstract Q value as the exploration constant. However, this technique has been improved by the dynamic, scale-independent exploration factor global-std [1]. The global-std exploration constant has the form $C \cdot \sigma$ where $\sigma$ is the standard deviation of the Q values of all nodes in the search tree and $C \in \mathbb{R}^+$ is some fixed parameter. Furthermore, we always use $K = 3$ as the recency counter, which was proposed by Anand et al. (2016).

**Problem models:** For this paper, we ran our experiments on a variety of MDPs, all of which are either from the International Probabilistic Planning Conference (Grzes et al., 2014) or are commonly used in the abstraction algorithm literature (Anand et al., 2015; 2016; Hostetler et al., 2015; Yoon et al., 2008; Jiang et al., 2014). All models were chosen such that they are not simultaneously sparse reward and deterministic, as in that case any intra-abstraction policy for the here-considered abstraction algorithm would have no effect at all. We ran all of our experiments on the finite-horizon versions of the considered MDPs with a default horizon of 50 steps and a discount factor $\gamma = 1$. If the reader is not familiar with any of the domains we used for the experiments, we provide a brief description for each MDP in the supplementary materials in Section A.12.

**Evaluation:** Each data point that we denote in the remaining sections of this paper (e.g. agent returns) is the average of at least 2000 runs. Whenever we denote a confidence interval for a data point, then this is always a confidence interval with a confidence level of 99% which is $\approx 2.33$ times the standard error. Furthermore, we use a borda-like ranking system to quantify agents' performances; in particular, we use *pairings* and *relative improvement scores*. For details, see supplementary Section A.6.

**Reproducibility:** For reproducibility, we released our implementation (Authors, 2025). Our code was compiled with g++ version 13.1.0 using the -O3 flag (i.e. aggressive optimization).

## 5 EXPERIMENTS

**Why intra-abstraction policies are needed:** For the first set of experiments, we validate that the case where intra-abstraction policies are required, i.e., two actions with the same parent but the same abstract node, is not an edge case but occurs frequently. Tab. 1 in the supplementary materials lists these statistics, showing that even for the least coarse abstraction setting, i.e., $\varepsilon_t = \varepsilon_a = 0$ and no partial grouping, there are a number of cases where an intra-abstraction policy has to be queried.

**Comparison of all intra-abstraction policies:** Next, we tested which of the intra-abstraction policies performs best overall by determining which policy the best-performing parameter combination

---

[1]Citation excluded for double anonymous review process

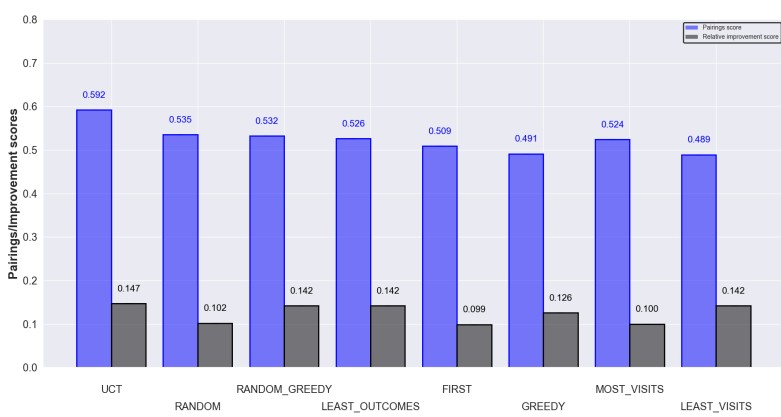

Figure 2: The pairings and relative improvement score for all iteration budgets combined of the best performing parameter-combination of each intra-abstraction policy are shown.

uses. In particular, we ran experiments with all intra-abstraction policies on pruned OGA, $(\varepsilon_a, \varepsilon_t)$-OGA, and RANDOM-OGA (OGA with random abstractions, see A.7, to test the behavior of the intra-abstraction policies on yet another abstraction type). For each abstraction algorithm, we varied $PG \in \{Yes, No\}$. For $(\varepsilon_a, \varepsilon_t)$-OGA, we tested $\varepsilon_a \in \{0, \infty\}$, $\varepsilon_t \in \{0, 0.2, 0.4, 0.8, 1.2, 1.6\}$, for pruned OGA we used $\alpha \in \{0, 0.1, 0.2, 0.5, 0.75, 1.0\}$, and for RANDOM-OGA we used $p_{abs} \in \{0.1, 0.2, 0.5, 1.0\}$. Each parameter combination was run with 100, 200, 500, and 1000 iterations using $C = 2$. The bar chart in Fig. 2 shows the pairings and relative improvement scores for the parameter combination with the highest scores for each intra-abstraction policy for all iteration budgets combined. The scores for each individual iteration budget are visualized in the supplementary materials in Fig. 5. The following key observations can be made.

1) First and foremost, the RANDOM intra abstraction policy that has thus far been implicitly used in the literature is always decisively beaten by at least one other strategy. Additionally, the FIRST strategy, which one might accidentally use when no proper tiebreaking is implemented, is even worse. This shows that choosing a suitable intra-abstraction policy is an important aspect when designing an abstraction algorithm.

2) Secondly, it is UCT that consistently performs either best or second best in both scores across all budgets, while all other methods fluctuate in their performance, which might be caused due to the few tasks they are calculated over (just a little over 10 environments). The overall best performing parameter-combination was $(0, \infty)$-OGA using $PG = no$ and UCT as the intra-abstraction policy (this achieved the values 0.592 and 0.147 in the all-budgets bar chart). The overall best RANDOM-using strategy in terms of the rel. improvement score was pruned OGA with $\alpha = 0.75, PG = no$ and pruned OGA with $\alpha = 0.5, PG = no$ for the pairings score (these correspond to the values 0.535 and 0.102 in the all-budgets bar chart).

3) For the remaining strategies, it is difficult to identify trends that are certainly beyond noise. However, given that RANDOM_GREEDY only slightly improves over RANDOM in the pairings score shows that, though the intra-abstraction policy at decision time has a considerable performance impact, the tree-policy intra-abstraction policy is equally as important as shown by the performance improvement of UCT.

In sum, the choice of an intra-abstraction policy can have a great impact on performance, which we further consolidate in the next subsection. While the best intra-abstraction policy depends on the concrete iteration budget - model setting, a consistent gain over RANDOM can be gained by simply replacing it with the UCT strategy. In the supplementary materials Section A.4, we show that this drop-in replacement improvement also holds when the abstractions themselves are fixed and one has to find a strategy that can best deal with these abstractions.

**Parameter-optimized performances:** Next, we compared RANDOM versus the UCT intra abstraction policy in the parameter-optimized setting, where we optimized both agents over the same set of parameters in addition to varying the exploration constant in $C \in \{0.5, 1, 2, 4, 8, 16\}$ and

including additional domain-specific $\varepsilon_a$ values that are listed in Tab. 3 in the supplementary materials. For feasibility reasons, we restricted ourselves to UCT instead of additionally including all other intra-abstraction policies. Fig. 3 shows the performance graphs for each environment. The following observations can be made.

1) In every environment UCT either clearly performs better than RANDOM with at least one iteration budget (in Crossing Traffic, Earth Observation, Manufacturer, Navigation, SysAdmin, Skill Teaching, Sailing Wind and Tamarisk) or performs on par (in Academic Advising, Game of Life, Cooperative Recon, Saving, Traffic, and Triangle Tireworld).

2) The performance gains can be explained by the fact that UCT performs better (relative to RANDOM), the coarser the abstraction, as shown in the supplementary materials Section A.9. The environments where performance is gained over RANDOM are those that satisfy the following criteria. Firstly, using coarse abstractions is either on par with OGA-UCT or even better. Secondly, the coarseness needs to introduce actual abstraction errors which for example, is rarely the case in Game of Life. Lastly, the intra-abstraction policy needs to be queried in the first place, which explains why there is no gain for Navigation at higher iteration budgets.

In summary, using intra-abstraction policies is a valuable tool to improve peak performances across a wide range of environments, especially when the peak performance without intra-abstraction policies has been reached with a coarse abstraction.

## 6 LIMITATIONS AND FUTURE WORK

In this paper, we first generalized the ASAP and AS frameworks to ASASAP. We then brought attention to the intra-abstraction policy problem and showed that this is not an edge case. To relieve this issue, we proposed several intra-abstraction policies as an alternative to the random policy that is implicitly used in standard OGA. While some of them were only marginally better than RANDOM, like MIN_VISITS and RANDOM_GREEDY, we found that UCT-OGA performs best and consistently, performing either on par or clearly outperforming standard OGA across a variety of parameter settings and environments. Consequently, we believe that UCT should be used as the standard intra-abstraction policy for MCTS-based abstraction methods instead of the random policy.

Firstly, limitations of OGA also directly translate to OGA enhanced with an intra-abstraction policy. In particular, for any performance gains to appear in the first place, the environment must contain state-action pairs with the same Q value. Furthermore, for any abstractions to be detected in the first place, the search graph must be a directed acyclic graph to ensure that there are state-action pairs with the same successors, a necessary condition for any abstractions (unless $\varepsilon_t = 2$). Another intra-abstraction policy-specific limitation is that these are only useful for non-exact abstractions. In particular, if only state-action pairs with the same optimal Q value are abstracted then which one the intra-abstraction policy chooses makes no difference.

From a different viewpoint, intra-abstraction policies can be seen as some special form of operating in hierarchical abstractions, where one iteratively selects actions from different layers of abstractions until a ground action is reached. In the case of intra-abstraction policies, the hierarchy consists of two abstractions: the standard ASAP abstraction, followed by a trivial one where every node is assigned to its own abstract node. We believe that choosing an intra-abstraction policy that itself just selects an abstract action from a finer abstraction could be worth investigating.

Whenever the intra-abstraction policies resulted in ties, these were always resolved randomly. Even though any gains here would be even more marginal, one could probably find further optimizations by setting up a tiebreak hierarchy, e.g., if UCT results in a tie, then these are resolved by MOST_VISITS.

Furthermore, on a more general level, it might be worth investigating if the progress in search abstractions can be translated to machine learning methods that are built on these searches, such as AlphaZero (Silver et al., 2017).

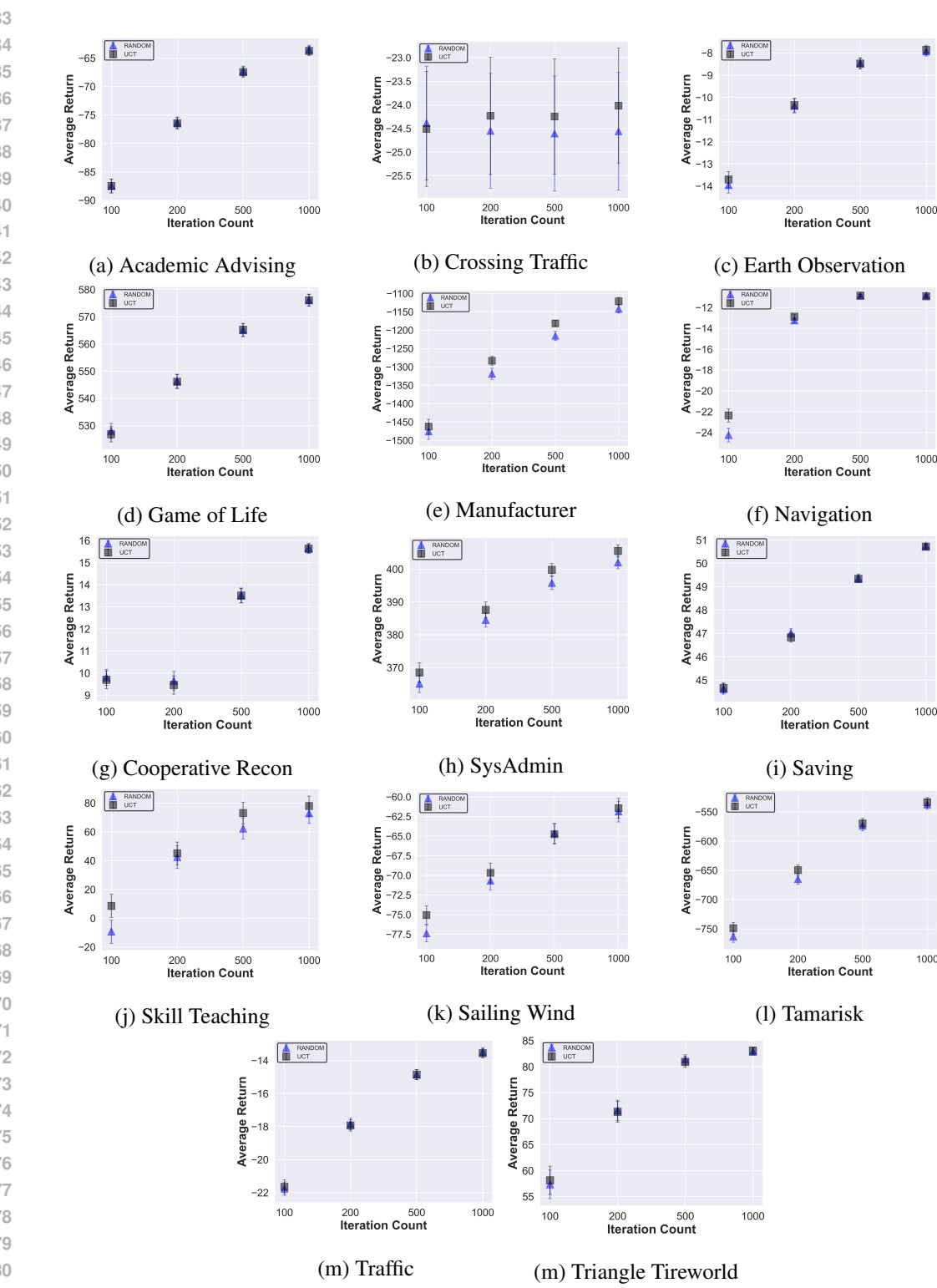

Figure 3: The performance graphs of in dependence of the MCTS iteration count of the parameter optimized versions of pruned OGA, $(\varepsilon_a, \varepsilon_t)$-OGA, and RANDOM-OGA combined with the UCT or RANDOM intra-abstraction policy.

## 7 REPRODUCIBILITY STATEMENT

In our experiment setup, we have a subsection called *Reproducibility* in which we provide a download link to the full codebase used for this project as well as compilation details. The codebase contains an elaborate README detailing the steps to reproduce the experiments.

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

## A  SUPPLEMENTARY MATERIALS

### A.1  PROOF OF THEOREM 1

Running the abstraction-using UCB MCTS variant on the local-layered MDP $M_l$ rooted at $s_d$ and the UCT intra-abstraction policy is equivalent to running standard MCTS on a move group MDP of $M_l$ (Eyck & Müller, 2011) where the move groups are the abstract actions. A move group MDP splits the original MDP's actions into two phases: the first is to select a move group (which is a subset of the available actions at the corresponding state), and then select an action within that move group that has the transition dynamics of the original MDP. The move-group selection action is deterministic and yields a reward of 0. Consequently, the $Q^*$ values of the original actions in the move group MDP have the same value as in the original MDP and therefore both MDPs have identical optimal policies (when using a discount of $\gamma = 1$ and excluding the move group actions). Since standard MCTS (using UCB) converges in probability to the optimal action (Kocsis & Szepesvári, 2006) on the move-group MDP, the same must hold for the here-considered abstraction-using MCTS variant. $\square$

### A.2  INTRA-ABSTRACTION POLICY QUERY STATISTICS

Table 1: Statistics of the ratio of tree policy calls where two actions of the same parent were part of the same abstract node to show that intra-abstraction policies are not an edge case. A ratio of $1.00$ means that an intra abstraction policy was always required, while a ratio of $0.00$ means that it was never required. The statistics were gathered with $(0, \varepsilon_t)$-OGA using the RANDOM intra abstraction policy, 1000 iterations, and $C = 4$. The results were averaged from 100 episodes each.

| Domain | $\varepsilon_t = 0$ NO-PG | PG | $\varepsilon_t = 0.8$ NO-PG | PG | $\varepsilon_t = 1.6$ NO-PG | PG |
|---|---|---|---|---|---|---|
| Academic Advising | 0.02 | 0.21 | 0.02 | 0.29 | 0.64 | 0.74 |
| Cooperative Recon | 0.30 | 0.33 | 0.33 | 0.32 | 0.28 | 0.32 |
| Crossing Traffic | 0.88 | 0.93 | 0.92 | 0.92 | 0.89 | 0.94 |
| Earth Observation | 0.00 | 0.00 | 0.27 | 0.27 | 0.27 | 0.27 |
| Game of Life | 0.00 | 0.00 | 0.52 | 0.84 | 0.84 | 0.82 |
| Manufacturer | 0.00 | 0.00 | 0.05 | 0.07 | 0.11 | 0.13 |
| Navigation | 0.00 | 0.00 | 0.01 | 0.02 | 0.02 | 0.13 |
| Racetrack | 0.17 | 0.51 | 0.17 | 0.51 | 0.17 | 0.51 |
| Sailing Wind | 0.00 | 0.02 | 0.00 | 0.06 | 0.05 | 0.13 |
| Saving | 0.00 | 0.00 | 0.00 | 0.01 | 0.01 | 0.01 |
| Skills Teaching | 0.01 | 0.03 | 0.11 | 0.11 | 0.13 | 0.20 |
| SysAdmin | 0.01 | 0.05 | 0.23 | 0.30 | 0.31 | 0.38 |
| Tamarisk | 0.00 | 0.01 | 0.26 | 0.34 | 0.40 | 0.44 |
| Traffic | 0.08 | 0.13 | 0.52 | 0.77 | 0.47 | 0.79 |
| Triangle Tireworld | 0.32 | 0.47 | 0.28 | 0.40 | 0.33 | 0.48 |

## A.3 ASAP ABSTRACTION EXAMPLE

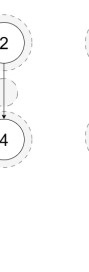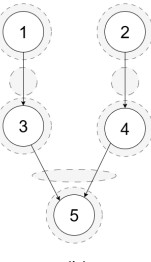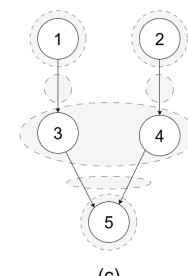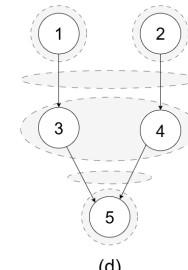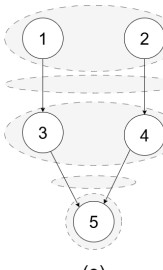

(a)   (b)   (c)   (d)   (e)

Figure 4: A showcase of how the ASAP abstraction framework, which itself is a special case of ASASAP abstractions, would detect equivalences in the following 5-state MDP. Each node represents a state, and arrows represent deterministic actions with the same immediate reward of 0. The dotted ovals represent abstractions. Initially, in (a), all states and state-action pairs are in their own singleton abstract node. Then, in (b) the next state-action pair abstraction is constructed (the application of function $f$ from Section 2) from this initial state abstraction, which groups the actions of nodes 3 and 4 because they have the same immediate reward and the same transition distribution. From this state-action pair abstraction the next state abstraction is constructed in (c), (the application of function $g$ from Section 2) which groups nodes 3 and 4 because they have the same set of abstract state-action pairs. Then again, in (d) the next state-action pair abstraction is constructed which also groups the actions from nodes 1 and 2 because they have the same abstract successor. Then a state abstraction is constructed again in (e), which groups states 1 and 2. Then further applications of $f$ or $g$ would have no effect, hence this abstraction is converged.

## A.4 PERFORMANCES ON FIXED ABSTRACTIONS

Next, it will be tested how well each intra-abstraction policy can generalize across different abstraction types. To do this, we reanalyzed the results of the main-part experiment section and built the pairings and relative improvement score when moving all parameters except the intra-abstraction to the set of tasks (i.e. a task now includes, for example, the $\varepsilon_t$ or $\alpha$ values). Tab. 2 shows both the relative improvement and pairings score for these results. Importantly, this shows that when confronted with an arbitrary abstraction, both FIRST and RANDOM perform worst by far, while UCT is by far the best policy.

Table 2: The relative improvement and pairings scores for each intra-abstraction policy, when for the score calculation all parameters except the intra-abstraction policy are part of the task set.

| Intra-abs policy | Rel. improv. score | Intra-abs policy | Pairings score |
|---|---|---|---|
| UCT | 0.074 | UCT | 0.394 |
| GREEDY | 0.046 | GREEDY | 0.206 |
| RANDOM_GREEDY | 0.040 | RANDOM_GREEDY | 0.102 |
| LEAST_OUTCOMES | 0.038 | LEAST_VISITS | 0.097 |
| LEAST_VISITS | 0.038 | LEAST_OUTCOMES | 0.077 |
| MOST_VISITS | −0.034 | MOST_VISITS | −0.145 |
| FIRST | −0.100 | RANDOM | −0.363 |
| RANDOM | −0.101 | FIRST | −0.368 |

## A.5 Domain-specific $\varepsilon_A$ values

Table 3: A list of the environment-specific $\varepsilon_a$ values that were used for the experiments. All domains that are not explicitly listed here use the default values $\varepsilon_a \in \{0, 1, 2, \infty\}$. The values were chosen to be equal to rewards (except 0 and $\infty$) that occur in these environments.

| Environment | $\varepsilon_a$ values |
|---|---|
| Academic Advising | $0, \infty$ |
| Cooperative Recon | $0, 0.5, 1.0, \infty$ |
| Crossing Traffic | $0, \infty$ |
| Manufacturer | $0, 10, 20, \infty$ |
| Skill Teaching | $0, 2, 3, \infty$ |
| Tamarisk | $0, 0.5, 1.0, \infty$ |
| Default | $0, 1, 2, \infty$ |

## A.6 Definition of the relative improvement and pairings score

In the main experimental section, we evaluated the intra-abstraction policies with respect to the relative improvement and pairings score, which are formalized here. While the pairings score is calculated by summing over the number of tasks where some agent performed better than another, the relative improvement score also takes the percentage of the improvement into account; however, it is prone to outliers. Hence, we considered both scores to paint the full picture.

Concretely, let $\{\pi_1, \ldots, \pi_n\}$ be $n$ agents (e.g., concrete parameter settings) where each agent was evaluated on $m$ tasks (e.g. a given MCTS iteration budget and an environment or a given abstraction in case of the experiments in Section A.4) where $p_{i,k} \in \mathbb{R}$ denotes the performance of agent $\pi_i$ on the $k$-th task.

**Definition:** The *pairings score matrix* $M \in \mathbb{R}^{n \times n}$ is defined as

$$M_{i,j} = \frac{1}{m-1} \sum_{1 \le k \le m} \mathrm{sgn}(p_{i,k} - p_{j,k}) \tag{7}$$

where sgn is the signum function. The *pairings score* $s_i \le i \le n$ is given by

$$s_i = \frac{1}{n-1} \sum_{1 \le l \le n, l \ne i} M_{i,l}. \tag{8}$$

**Definition** The *relative improvement matrix* $M \in \mathbb{R}^{n \times n}$ is defined as

$$M_{i,j} = \frac{1}{m-1} \sum_{1 \le k \le m} \frac{p_{i,k} - p_{j,k}}{\max(|p_{i,j}|, |p_{j,k}|)} \tag{9}$$

and the *relative improvement score* $s_i \le i \le n$ is given by

$$s_i = \frac{1}{n-1} \sum_{1 \le l \le n, l \ne i} M_{i,l}. \tag{10}$$

## A.7 RANDOM-OGA

OGA that uses random abstractions is called RANDOM-OGA and functions as follows. Whenever a Q node Q is visited for the K-th time and its current abstract node consists only of itself, then with the probability $p_{\mathrm{abs}} \in [0, 1]$, Q's abstract node is changed with uniform probability to any of the abstract nodes of the same depth. Initially, at creation, any Q node is its own abstract node. Note that RANDOM-OGA does not abstract states, as it is only state-action pair abstractions that influence the agent's decision-making through the modified UCB formula.

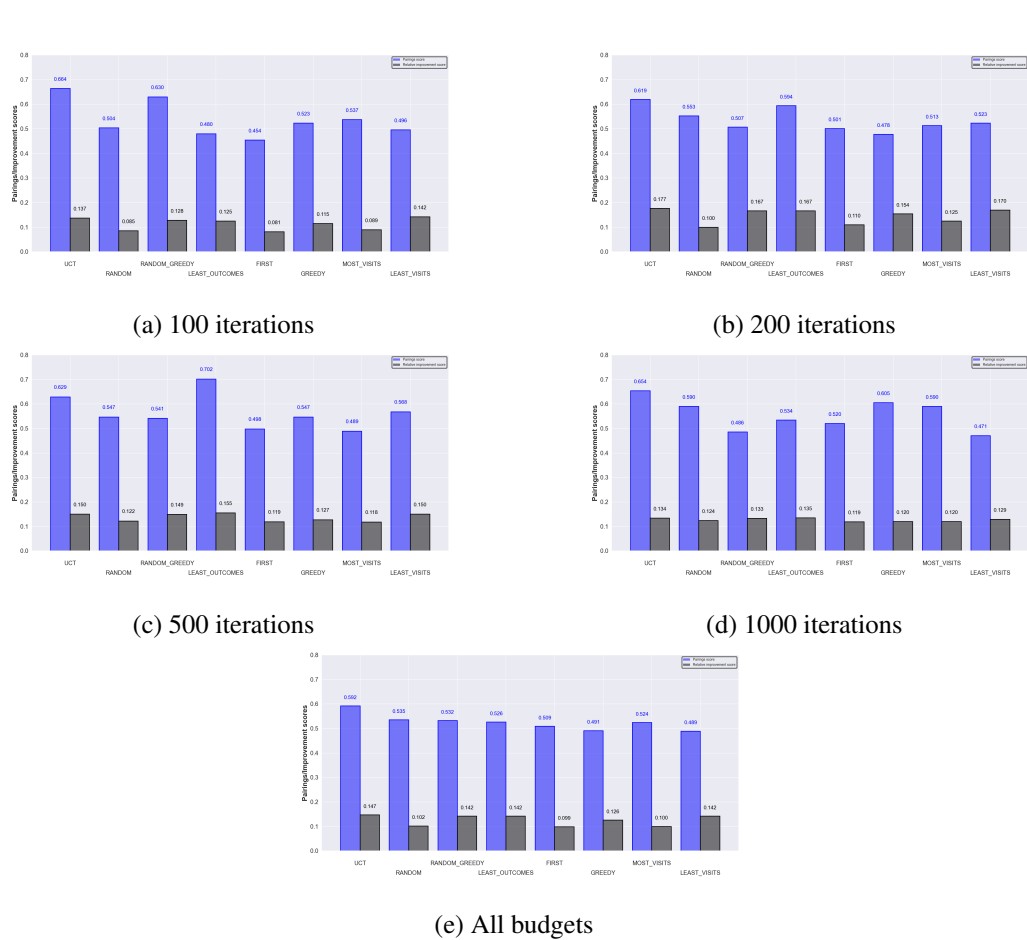

(a) 100 iterations

(b) 200 iterations

(c) 500 iterations

(d) 1000 iterations

(e) All budgets

Figure 5: For each iteration budget and the combination of all iteration budgets, the pairings and relative improvement score for the best performing parameter-combination of each intra-abstraction policy are shown.

## A.8    PAIRINGS AND RELATIVE IMPROVEMENT SCORES FOR EACH INDIVIDUAL ITERATION BUDGET:

## A.9    ABLATION: PERFORMANCES WITH VARYING ABSTRACTION COARSENESSES

Lastly, we conducted an ablation on the efficiency of UCT and RANDOM as the intra-abstraction policy in dependence on the coarseness of the abstraction. Concretely, we reanalyzed the pairings and relative improvement scores from the experiment section that were created over all parameter combinations. Fig. 6 shows the scores of several abstractions with varying coarsenesses and the highest pairings/relative improvement score that UCT and RANDOM could achieve in that setting. The results are pretty clear: The coarser the abstraction, the greater the performance gap between UCT and RANDOM. Interestingly, using UCT can change the location of the performance peaks by enabling coarser abstractions: In $(\varepsilon_a, \varepsilon_t)$ the peak is obtained at $\varepsilon_t = 0.4$ instead of $\varepsilon_t = 0.2$, and in pruned OGA the peak is obtained at $\alpha = 1$ instead of $\alpha = 0.75$.

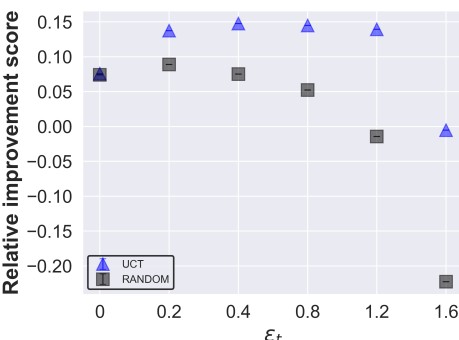

(a) Relative improvement score on $(\varepsilon_{\mathrm{a}}, \varepsilon_{\mathrm{t}})$-OGA abstractions.

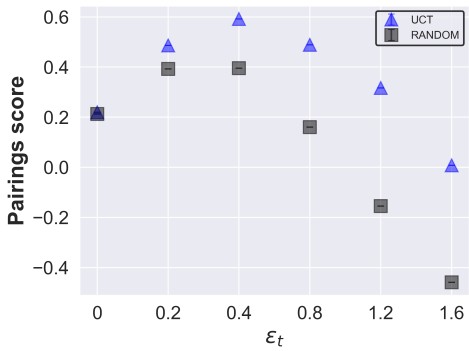

(b) Pairings score on $(\varepsilon_{\mathrm{a}}, \varepsilon_{\mathrm{t}})$-OGA abstractions.

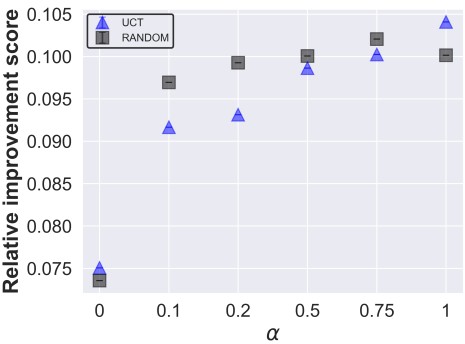

(c) Relative improvement score on pruned OGA abstractions.

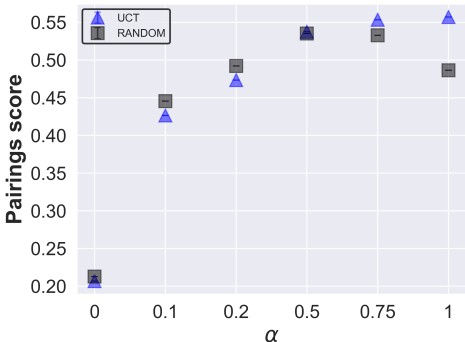

(d) Pairings score on pruned OGA abstractions.

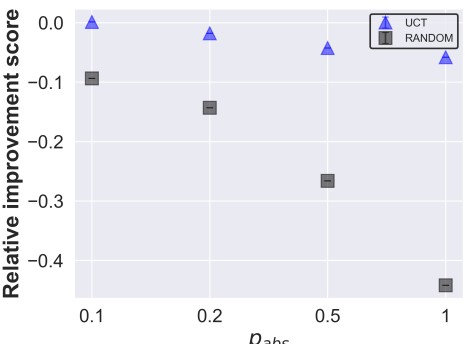

(f) Relative improvement score on RANDOM-OGA abstractions.

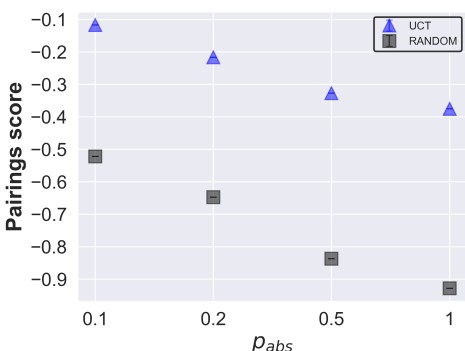

(e) Pairings score on RANDOM-OGA abstractions.

Figure 6: For RANDOM-OGA, pruned OGA, and $(\varepsilon_{\mathrm{a}}, \varepsilon_{\mathrm{t}})$-OGA, these figures show the parameter combinations using UCT and RANDOM as the intra-abstraction policies and the respective abstraction parameter (e.g. $\alpha = 1$ that achieved the highest pairings or relative improvement score from the main-part experiment section.

## A.10 MONTE CARLO TREE SEARCH

All here-presented abstraction algorithms rely on Monte Carlo Tree Search (MCTS) which we are going to describe now. Let $M$ be a finite-horizon MDP. On a high level, MCTS repeatedly samples trajectories starting at some state $s_0 \in S$ where a decision has to be made until a stopping criterion is met. The final decision is then chosen as the action at $s_0$ with the highest average return. In contrast to a pure Monte Carlo search, MCTS improves subsequent trajectories by building a tree (or, in our

case, a directed acyclic graph) from a subset of the states encountered in the last iterations, which is then exploited. In contrast to pure Monte Carlo search, MCTS is guaranteed to converge to the optimal action.

An MCTS directed acyclic graph is made of two components. Firstly, the state nodes, that represent states and Q nodes that represent state action pairs. Each state node, saves only its children which are a set of Q nodes. Q nodes save both its children which are state nodes and the number of and the sum of the returns of all trajectories that were sampled starting at the Q node.

Initially, the MCTS search graph consists only of a single state node representing $s_0$. Until the iteration budget is exhausted, the following steps are repeated.

1. **Selection phase**: Starting at the root node, MCTS first selects a Q node according to the so-called *tree policy*, which may use the nodes' statistics, and then samples one of the Q node's successor states. If either a terminal state node, a state node with at least one non-visited action (partially expanded), or a new Q node successor state is sampled that is not represented by another node of the same layer, the selection phase ends.

   A commonly used tree policy (**and the one we used**) that is synonymously used with MCTS is Upper Confidence Trees (UCT) (Kocsis & Szepesvári, 2006), which selects an action that maximizes the Upper Confidence Bound (UCB) value. Let $s \in S$ and $V_a, N_a$ with $a \in \mathbb{N}$ be the return sum and visits and of the Q nodes of the node representing $s$. The UCB value of any action $a$ is then given by

$$\text{UCB}(a) = \underbrace{\frac{V_a}{N_a}}_{\text{Q term}} + \underbrace{\lambda \sqrt{\frac{\log\left(\sum_{a' \in \mathbb{A}(s)} N_{a'}\right)}{N_a}}}_{\text{Exploration term}} . \tag{11}$$

   The exploration term quantifies how much the Q term could be improved if this Q node was fully exploited and is controlled by the exploration constant $\lambda \in \mathbb{R} \cup \{\infty\}$. If one chose $\lambda = 0$, the UCT selection policy becomes the greedy policy and for $\lambda = \infty$, the selection policy becomes a uniform policy over the visits. In case of equality, some tiebreak rule has to be selected, which is typically a random tiebreak. From here, will use MCTS and UCT (MCTS with UCB selection formula) synonymously.

2. **Expansion**: Unless the selection phases ended in a terminal state node, the search directed acyclic graph is expanded by a single node. In case the selection phase ended in a partially expanded state node, then one unexpanded action is selected (e.g. randomly, or according to some rule), the corresponding Q node is created and added as a child and one successor state of that Q node is sampled and added as a child to the new Q node. If the selection phase ended because a new successor of a Q node was sampled, then a state node representing this new state is added as a child to that Q node.

3. **Rollout/Simulation phase**: Starting at the state $s_{rollout}$ of the newly added state node of the expansion phase (or at a terminal state node reached by the selection phase), actions according to the *rollout policy* are repeatedly selected and applied to $s_{rollout}$ until a terminal state is reached. All states encountered during this phase are not added to the search graph.

4. **Backpropagation**: In this phase, the statistics of all Q nodes that were part of the last sampled trajectory that corresponds to a path in the search graph are updated by incrementing their visit count and adding the trajectory's return (of the trajectory starting at the respective Q node) to their return sum statistic.

## A.11 RUNTIME MEASUREMENTS

Tab. 4 lists the average decision-making times for each environment of the UCT intra-abstraction policy compared to RANDOM for 100 and 2000 iterations on states sampled from a distribution induced by random walks. This shows that while UCT adds only a minor overhead, despite having to execute more UCB evaluations. In particular, we are using highly optimized environment implementations that could be the runtime bottleneck in more complex environments.

Table 4: Average decision-making times of MCTS using either the UCT or RANDOM intra-abstraction policy in milliseconds for 100 and 2000 iterations using $\varepsilon_t = 0.8$. This data was obtained using an Intel(R) Core(TM) i5-9600K CPU @ 3.70GHz. The data shows a median runtime overhead of $\approx 0.6\%$ for 100 iterations and $\approx 2\%$ for 2000 iterations.

| Domain | UCT-100 | RANDOM-100 | UCT-2000 | RANDOM-2000 |
|---|---|---|---|---|
| Academic Advising | 2.10 | 1.98 | 109.12 | 100.90 |
| Cooperative Recon | 3.67 | 3.62 | 182.63 | 179.36 |
| Crossing Traffic | 2.33 | 2.32 | 346.46 | 343.24 |
| Earth Observation | 7.38 | 7.35 | 288.69 | 290.51 |
| Game of Life | 3.50 | 3.54 | 149.85 | 140.11 |
| Manufacturer | 10.05 | 10.04 | 271.58 | 267.35 |
| Navigation | 2.29 | 2.20 | 61.59 | 57.69 |
| Sailing Wind | 2.11 | 2.12 | 144.28 | 142.40 |
| Saving | 1.34 | 1.37 | 70.02 | 68.03 |
| Skills Teaching | 3.55 | 3.52 | 179.40 | 175.79 |
| SysAdmin | 1.56 | 1.51 | 101.27 | 114.00 |
| Tamarisk | 2.81 | 3.06 | 126.52 | 117.40 |
| Traffic | 3.62 | 3.57 | 114.41 | 112.90 |
| Triangle Tireworld | 4.01 | 3.65 | 119.80 | 108.21 |

## A.12 PROBLEM DESCRIPTIONS

We will provide a brief description of each of the IPPC problems used for the experiments.

- **Academic Advising**: The Academic Advising domain was used for the IPPC 2014 (Grzes et al., 2014). The agent is a student whose goal is to pass certain academic classes. Formally, the state is an element in $\{P, \text{NP}, \text{NT}\}^n$ (representing for each course whether it has been passed, not been passed, or not been taken), and the agent's action is to choose a course to take. The course outcome depends on the states of the prerequisite courses. The episode ends when all courses are passed, and while not all mandatory courses, a subset of all courses, are passed, the agent incurs a constant penalty per step.

- **CooperativeRecon**: This domain models a robot having to prove the existence of life on a foreign planet. The robot is modeled as moving on a 2-dimensional grid which contains a number of objects of interest and a base. If the agent is at an object of interest, it can survey the object for the existence of water and life. The probability of a positive result of the latter is dependent on whether water has been detected. If life has been detected, the agent may photograph the object of interest which is the only way to gain a reward. Each detector may break on usage making it either unusable or decreasing its chance of working. The detectors can be repaired at the base.

- **Crossing Traffic**: Crossing traffic is a grid-based navigation problem where the agent has to maneuver through lanes of traffic. Obstacles can move only on the x-axis from right to left. Obstacles spawn randomly on the right end of the grid except for the first and last row of the grid and on collision with the agent, render the agent unable to move, thus making the current episode impossible to be solved. Starting at the top left of the grid, the goal is to reach the bottom left grid cell in as few steps as possible without getting hit.

- **EarthObservation**: EarthObservation was a test problem for the IPPC 2018 which models a satellite orbiting earth. Formally, each state is a position on a 2-dimensional grid, representing the satellite's longitudinal position and the latitude the camera is aimed at as well as weather levels for some designated cells. At each step, the weather levels stochastically change independent of the agent's actions which are to idle, to take a photo of the current position, or increment/decrement the current cells $y$-position (i.e. shifting the camera focus). A reward is obtained if one of the designated cells is photographed with an amount depending on the cell's current weather condition.

- **Game of Life**: The original game of life by John Conway (Gardner, 1970) is a cellular automaton and modified into a stochastic MDP as a test problem for the International Probabilistic Planning Competition (Sanner & Yoon, 2011) by introducing noise to the

deterministic state transition, setting the current number of alive cells as the reward, and allowing the agent to choose one cell which will contain a living cell with a high probability. States are elements in $\{0, 1\}^{n \times n}$ describing whether there is an alive cell at each cell on a grid. To reduce the action space that scales quadratically which the grid length, we allow only a subset of the original actions, which is to specify one alive cell that is prevented from dying.

- **Manufacturer**: In this domain, the agent manages a manufacturing company. The agent's ultimate goal is to sell goods to customers. However, to sell a good, the agent has to first produce the good, which may require building factories and acquiring the necessary goods required for production. Additional difficulty comes from the fact that the goods' price levels vary stochastically.

- **Navigation**: Navigation was a test problem for the International Probabilistic Planning Competition 2011 (Sanner & Yoon, 2011). The goal is to move a robot on an $n \times m$ grid from $(n, 1)$ to $(n, m)$ in the least number of steps. The robot may move to any of the four adjacent tiles, however, each tile is assigned a unique probability with which the robot is reset back to $(n, 1)$. At each step, except the one where the goal is reached, the agent incurs a constant negative reward, making the objective to reach the goal state as quickly as possible.

  **Saving**: Saving is introduced by Hostetler et al. (2015), where the agent aims to maximize accumulated wealth over time. At each step, the agent can choose one of three actions: Invest, Borrow, or Save. Borrow provides an immediate reward of 2 but imposes a penalty of -3 after $n$ time steps. Once this action is taken, it cannot be repeated until the delayed penalty is applied. Save yields an immediate reward of 1 with no further consequences. Invest offers no immediate reward but enables the agent to take the Sell action within the next $m$ time steps. The agent cannot invest again until either the Sell action is executed or $m$ steps have elapsed. If Sell is chosen, then the agent receives a reward equal to the current price level that changes stochastically and independently of the agent's actions.

- **Sailing Wind**: Originally proposed by Robert Vanderbei (Vanderbei, 1996), the goal of Sailing Wind is to move a ship that starts at $(1, 1)$ on an $n \times n$ grid to $(n, n)$ with minimal cost. There is no consistent use of a transition and reward function throughout the literature. There may just be two available actions (*down*, *right*) (Jiang et al., 2014) or up to seven (each adjacent cell except the one facing a stochastic wind direction) (Anand et al., 2015). The cost of each action is dependent on the current wind direction which stochastically changes its direction at each step independent of the player's actions.

- **SysAdmin**: Used as a test problem for the IPPC 2011, a SysAdmin instance is a graph (describing a network topology) with $n \in \mathbb{N}$ vertices. The state space is $\{0, 1\}^n$ (describing which machines are currently operating) and the action space is $\{1, \ldots, n\}$ (describing with machine to reboot). At each step, the reward is dependent on the machines that are currently working, a reboot causes the rebooted machine to have a high chance of working in the next step. Machines can randomly fail at each step, however this probability is increased when a neighbor fails.

- **Tamarisk**: Tamarisk is yet another problem from the IPPC 2014 (Grzes et al., 2014) which models the expansion of an invasive plant in a river system. The river system is modelled as a chain of reaches where each reach contains a number of slots that may be unoccupied, occupied by a native plant, or occupied by the invasive Tamarisk plant. Both plant types spread stochastically to neighboring states with a higher probability of spreading downstream. At each time step, the agent chooses an action for one reach, which are doing nothing, eradicating Tamarisk, or restoring a native plant. The action chosen at a reach is applied to all slots in that reach. Except for the do-nothing action, all actions can randomly fail. The agent has to balance the action's costs with the penalties incurred for existing Tamarisk plants.

- **Triangle Tireworld:** Tireworld was proposed as a test problem for the IPPC 2004 (Younes et al., 2005). In the original goal-based version, the agent is a car that traverses a graph. At each step, the car may move to an adjacent node, change its tire, or load a tire. The goal is to reach a designated goal node. At each step, the car's tire may randomly break. If the car isn't carrying a spare tire, the goal can no longer be reached. Otherwise, if available, a

spare tire (at most one can be carried) must be used to replace the current tire. Some nodes contain spare tires, which when the agent visits them, can be picked up.

- **Skills Teaching**: This domain models a student-teacher interaction, where the agent plays the role of the teacher. There is a fixed number of skills that form a directed graph of prerequisites. The student possesses one of three levels of sufficiency at each skill. The agent is rewarded for each skill being at the highest sufficiency and punished for each skill at the lowest sufficiency level. At each, step the agent may choose a skill for which to pose a question to the student or give the student a direct hint. The student can increase their sufficiency at that skill for correctly answering a question and lose sufficiency for answering wrong. The probability of getting a question right is dependent on the sufficiency of the skill's prerequisite. A hint can elevate the student to the medium sufficiency level directly but only if all prerequisites are at the highest sufficiency.

- **Traffic**: This problem models a traffic system in which the agent is tasked with controlling/advancing intersections with the goal of minimizing congestion. The traffic system is modeled as a directed graph and each vertex is either empty or occupied. Occupancy flows along the graph's edges except for some designated intersection edges where the flow is dependent on the intersection's state. The only stochasticity of this MDP arises in the form of cars spawning randomly at the designated perimeter vertices. The agent receives a reward equal to the negative number of occupied vertices that have one predecessor vertex that is also occupied.

Constrictor is played on an $n$ times $n$ grid. Players take turns moving to any of the neighboring (4-neighborhood) grid cells that neither moves the player out of bounds nor hits any cell that has already been visited by any of the two players. The game ends when one player has nowhere left to move.

