# OpenReview forum: "Investigating intra-abstraction policies for non-exact abstraction algorithms"
_ICLR.cc/2026/Conference — ICLR 2026 Conference Withdrawn Submission_

### Official Review · Reviewer_BJ4r · 2025-10-31

**Soundness:** 2
**Presentation:** 1
**Contribution:** 2
**Rating:** 4
**Confidence:** 2

**Summary:**

The authors propose a MCTS variant algorithm where state-action space is employed to help the MCTS run more efficiently. State-action abstraction is derived from the deviation in reward function and transition function for any two state action pairs - if sufficiently close enough then the state-action pair can be grouped together. Some weak theoretical guarantees are provided with respect to convergence in the root node. An empirical analysis is conducted to on two types of MCTS, RANDOM and UCT, demonstrating that RANDOM selection policy is superior to UCT in the state-action abstracted MCTS.

**Strengths:**

- This paper addresses a key problem in making efficient MCTS iterations, and the ideas behind state-action abstraction are sound.

- There are a wealth of environments (academic advising, Manufacturer etc.) in which the MCTS variants are subject to.

**Weaknesses:**

- The paper is quite difficult to parse, and the paper over requires a large amount of revision in order for its ideas to be clearly communicated. Based on this - I perhaps did not understand the paper that well. For example, for readers not super familiar with previous work in state-action abstraction, a short formal presentation of ASAP, ASASAP, and what the author's contribution is, would be nice to have - and not too complex to communicate.

- Lack of theoretical depth. The paper contains very little theoretical analysis and relies on its empirical arguments to convince the reader. (Which is a weakness if the empirical evidence is insufficient).

- Insufficient empirical analysis. Given that there is a dearth of theoretical content in the paper, the empirical evidence is also somewhat not convincing. Only two flavours of MCTS are considered in the empirical analysis RANDOM and UCT, but many other selection strategies are considered in lines 219 - 240 are not accounted for.

**Questions:**

- There seems to be just one case study UCT vs RANDOM in the empirical results of the main paper, should we not have more variants being of MCTS being compared?

- What is the goal of Theorem 1? It seems to me that it tells us that policy at the root node will have a optimal action probability converging to 1. To be specific are we referring to the optimal real action leading to the real state, or the optimal abstraction of the root action.

- To follow up on Theorem 1, would this apply to all nodes, or only specifically the root node? And is there any intuition behind this.

- Consider Eq. 5 as the state-action abstraction criteria. Could we not say that if \epsilon becomes big enough, all states and actions can be grouped together. We would then consider this a bandit problem then correct? What is the trade-off here? Intuitively I would assume that as we start group state-action pairs together, we lose expressiveness of the policy, which could lead to higher variance in the final reward trajectory or variance in policy, is this considered in the paper's ablation study or theoretical analysis?

- w.r.t. to the environments, is there any information about how big the state action space is, and how the abstraction reduced the search space (and the corresponding trade-offs)?

---

### Official Review · Reviewer_ZS21 · 2025-10-31

**Soundness:** 3
**Presentation:** 3
**Contribution:** 3
**Rating:** 6
**Confidence:** 3

**Summary:**

This paper addresses an issue in those Monte Carlo Tree Search (MCTS) methods that use abstraction algorithms to reduce the search space, such as pruned On the Go Abstractions (pruned OGA). The usage of abstraction algorithms can group multiple actions from the same parent state into a single abstract node. This results in all actions within that group having identical Upper Confidence Bound (UCB) values, necessitating a tiebreak rule. But this rule in the pruned OGA has implicitly been random. The paper introduces the Alternating State And State-Action Pair Abstractions (ASASAP) framework to generalize and unify existing abstraction methods. Based on this framework, the authors propose and empirically evaluate seven alternative "intra-abstraction policies", demonstrating that this tiebreaking scenario is a frequent occurrence and not an edge case. Their experiments show that one of the seven policies, the UCT policy, consistently outperforms the implicitly used random policy across various environments and parameter settings, serving as a parameter-free, drop-in improvement.

**Strengths:**

- This paper identifies a previously overlooked issue in state-of-the-art abstraction algorithms for MCTS. It successfully argues that the implicit use of a random tiebreak rule is a significant weakness.
- This paper introduces the ASASAP framework to unify different MCTS-based abstraction algorithms.
- The proposed policy, UCT, is a "parameter-free drop-in improvement" to replace the random policy used in the OGA. It is a practical, easy-to-implement approach for researchers in this area.

**Weaknesses:**

- The paper introduces the ASASAP framework and then proposes different polices. But it is not clear how to get those police from the framework, or how the ASASAP framework provides new insights into the intra-abstraction policy problem.
- In the experimental setting, some parameters are fixed. There is no discussion about how those parameters affect the results. It would be beneficial to show results with at least one alternative.

**Questions:**

Please refer to the Weaknesses.

---

### Official Review · Reviewer_3AvS · 2025-11-02

**Soundness:** 1
**Presentation:** 1
**Contribution:** 1
**Rating:** 0
**Confidence:** 4

**Summary:**

State abstractions are used to reduce the search complexity of Monte Carlo tree search by aggretating states, actions, or state-action pairs into abstract states and actions. Search is then run over this abstracted space which is typically much smaller than the original space. This paper investigates different approaches to select actions within abstracted action spaces.

**Strengths:**

-  The choice of evaluation domains and the number of seeds over which experiments were run seem appropriate.

**Weaknesses:**

- The paper has negligible significance.
    - The first purported contribution is a framework generalizing the formalism of abstractions used for MCTS. This is not a novel contribution. Their frameworks appears to be simply restating bisimilarity and bisumlation metrics. Furthermore, they do not actually use their framework for anything --- they describe it but then don't show how it's useful, or what insights can be gleaned using it, nor do they use it for the rest of paper.
    - The second half of the paper is about how actions can be selected from within an abstracted action. That is, when a number of actions are abstracted away, and that abstracted action is selected, how does one selected one of the constituent actions. The paper looks at several selection methods to do so. This is also not a significant contribution. The authors do not propose any new selection policy. The authors conclude that UCT is a better selection method than random, greedy, or least visits. However, the results do not support this claim (see further).

- The results do not support the conclusions made by the author. Figure 2 lacks any confidence intervals over the expected results. The performance of the different policies are close enough that without any confidence intervals, I cannot tell if any of the differences are significant. The authors also claim that Figure 3 shows that UCT performs better than random. This is completely false. In all the environments, over all the iterations, UCT is significantly better than random on only two points. Otherwise, there is no difference in performance.

- The paper is poorly written. There are several occasions the authors use notations and terms without defining them. Section 2 is difficult to understand.

**Questions:**

No questions.

**Details Of Ethics Concerns:**

This is one in a set of papers I am reviewing where large swathes of text are identical to each other. It is clear the authors submitted several papers, that all share the same sections almost completely word-for-word.

---

> ### Author Response · Authors · 2025-11-18
> **Addressing ethical concerns**
>
> Due to double-blind review, we can neither confirm nor reject the claim that the reviewer may have received multiple of our works for review. In fact, we submitted 4 papers to this conference. Those do not represent a dual-submission but independent algorithmic developments based on a unified benchmark. Each of our submitted papers are in the general area of abstraction-based search. However, each submission studies a different algorithmic approach and makes its own primary technical and empirical contributions warranting a unique submission. These algorithms, their theoretical analyses, and their empirical results are non-overlapping. Each paper can be read and evaluated independently.
>
> To the best of our understanding, the conference allows multiple submissions from the same group as long as they present distinct, non-redundant scientific contributions, which is exactly how we structured these papers.
>
> Taking the reviewers concerns seriously, we still decided to withdraw our submission to make sure that even such minor similar sounding passages will be restructured before resubmission to this or another conference.

---

### Note · Authors · 2025-11-18

I have read and agree with the venue's withdrawal policy on behalf of myself and my co-authors.